# A Novel Hybrid Algorithm for the Forward Kinematics Problem of 6 DOF Based on Neural Networks

**DOI:** 10.3390/s22145318

**Published:** 2022-07-16

**Authors:** Huizhi Zhu, Wenxia Xu, Baocheng Yu, Feng Ding, Lei Cheng, Jian Huang

**Affiliations:** 1Engineering Research Center for Intelligent Production Line Equipment of Hubei Province, School of Computer Science and Engineering Artificial Intelligence, Wuhan Institute of Technology, Wuhan 430205, China; zhuhuizhi@stu.wit.edu.cn (H.Z.); 11020501@wit.edu.cn (B.Y.); 2School of Computer Science, South-Central Minzu University, Wuhan 430074, China; dingfeng@mail.scuec.edu.cn; 3School of Information Science and Engineering, Wuhan University of Science and Technology, Wuhan 430081, China; chenglei@wust.edu.cn; 4Key Laboratory of Image Processing and Intelligent Control, School of Automation, Huazhong University of Science and Technology, Wuhan 430074, China; huang_jan@mail.hust.edu.cn

**Keywords:** Gough–Stewart, forward kinematics problem, ABC–BPNN, Newton’s method

## Abstract

The closed kinematic structure of Gough–Stewart platforms causes the kinematic control problem, particularly forward kinematics. In the traditional hybrid algorithm (backpropagation neural network and Newton–Raphson), it is difficult for the neural network part to train different datasets, causing training errors. Moreover, the Newton–Raphson method is unable to operate on a singular Jacobian matrix. In this study, in order to solve the forward kinematics problem of Gough–Stewart platforms, a new hybrid algorithm is proposed based on the combination of an artificial bee colony (ABC)–optimized BP neural network (ABC–BPNN) and a numerical algorithm. ABC greatly improves the prediction ability of neural networks and can provide a superb initial value to numerical algorithms. In the design of numerical algorithms, a modification of Newton’s method (QMn-M) is introduced to solve the problem that the traditional algorithm model cannot be solved when it is trapped in singular matrix. Results show that the maximal improvement in ABC–BPNN error optimization was 46.3%, while the RMSE index decreased by 42.1%. Experiments showed the feasibility of QMn-M in solving singular matrix data, while the percentage improvement in performance for the average number of iterations and required time was 14.4% and 13.9%, respectively.

## 1. Introduction

Gough–Stewart platforms (GSPs) are typical six-degree-of-freedom (6-DOF) parallel robots that have the advantages of high rigidity, high precision, and large carrying capacity [1]. GSPs are widely used in various large-scale motion simulation platforms such as flight, automotive, ship, and tank simulators. Generally, GSPs comprise three structures: universal and cylindrical and universal joint (6-UCU), 6-UPS, and 6-SPS, in which 6 represents six identical structures, U represents a universal joint, C represents a cylindrical joint, P represents a prismatic joint, and S represents a spherical joint. GSPs can be divided into 6–6, 3–3, and 6–3 structures according to the connection modes of the upper and lower hinge points. In this paper, the 6-UCU structure and 6–6 connection model are used. GSPs, with its structure and benefits, compensates for the many deficiencies of serial robots. It has a wide variety of applications in the future industry and intelligent manufacturing.

Because it can give mapping between a Cartesian and joint space, the kinematics problem is critical for parallel robotics. Solving the forward kinematics problem (FKP) is an essential step in the modeling and control of parallel robots, particularly for real-time applications. The FKP is a challenging and essential robotics topic in GSPs. Due to the high nonlinearity and the varied closed-loop kinematic architectures of parallel robots [2], there is currently no acknowledged generic solution to solve the FKP. To control GSPs, advanced algorithms with higher computation loads that achieve better performance can be used. Therefore, it is significant to find a more powerful algorithm to reduce the computation time of the FKP. The FKP of GSPs is, therefore, a hot topic for researchers.

Studies on the FKP of GSPs can be classified into the traditional approach and the intelligent algorithm approach. Conventional traditional approaches include the numerical method, analytical method, and adding extra sensors. The numerical method can obtain an iterative solution without suffering the multisolution problem [3]. The Newton–Raphson (NR) method is a prominent numerical approach for analyzing a parallel robot’s forward kinematics [4]. The NR method with good initial values is used to numerically solve nonlinear FKP equations for root finding. The method is overly sensitive to initial values, and results diverge if the initial values are inappropriate. The analytical method tends to find a closed-form solution [5,6]. This method forms a series of sophisticated derivations and is only useful for certain structures. The method of adding more sensors [7] is a strategy to find a unique approach with the least amount of computing. Unfortunately, due to the high cost of this method, and measurement and assembly faults, its application range is limited.

With the advancement of computer application technology and artificial intelligence, an increasing number of researchers have begun to use intelligent algorithms to solve the FKP, such as artificial neural networks (ANNs) and support vector machines (SVM). Morell et al. [8] solved FKP by using the support vector regression approach, which had a unique notion, but the model training time was too long and would not completely satisfy the requirements in real-time control. ANNs are employed in the field of intelligent algorithms to train the inverse kinematics values of the GSP to produce a set of forward kinematics solutions [9]. Subsequently, the NR method is used to obtain the exact values of the approximations. Using the hybrid strategy to solve FKP has been recognized by some scholars. This hybrid strategy [10] can be combined with a high level of solving speed and accuracy. Moreover, it can leverage the superb initial value provided by a neural network to perform a Newtonian iteration, which compensates for the Newtonian iteration’s sensitivity to the initial value. It also overcomes the problem of a neural network’s predicted value having a forecasting deviation. The approach is adaptable to parallel robots with a variety of structural properties [11]. Many researchers have used this foundation to improve the algorithm’s efficiency and accuracy in real-time motion. To ensure the feasibility of real-time application, Zhu et al. exploited the optimization of numerical iteration efficiency by creating a deviation-driven algorithm [12]. After that, a velocity Jacobian matrix iteration was used to improve the hybrid algorithm’s overall speed [13]. The advantages of the neural network and genetic algorithm are combined in the neurogenetic algorithm [14], which maximizes the robot’s workspace volume.

For the neural network part, the well-known BP neural network (BPNN) is mostly employed in the traditional hybrid strategy. BPNN has good nonlinear mapping ability, i.e., $f\left\{L_{O1},L_{O2},L_{O3},L_{O4},L_{O5},L_{O6}\right\}\to \left\{\alpha ,\beta ,\gamma ,x,y,z\right\}$, which is the relationship between joint displacement and end-effector pose. Although the BPNN model can achieve promising performance and has a flexible network structure, it still has some drawbacks: (i) For BPNNs, the training effect depends on the dataset, i.e., limited self-adaptation ability for different datasets. (ii) It is possible to fall into a local optimum for the model system with six inputs and six outputs, which cannot ensure a global training effect. (iii) The model is sensitive to initial parameters. Randomly initializing weights and thresholds impacts the model. Moreover, there is also a flaw with the numerical algorithm part. (iv) A set of inverse of Jacobian matrices $J^{-1}$ are formed in the process of solving nonlinear equations with the Newton–Raphson method [12]. That is, when the Jacobian matrix is singular, the equations have no solution. On the basis of the above problems, this paper proposes a novel hybrid algorithm that can guarantee efficient and accurate problem solving, so that any value input to the mechanism under the feasible motion space can have a solution, rendering it a universal algorithm for the 6-DOF forward kinematics problem.

The major contributions of our work are as follows:An optimization method via an artificial bee colony (ABC) to optimize the BPNN (ABC–BPNN) is proposed. Good weights and thresholds are obtained through the process of the ABC’s population iteration to prevent the training model from falling into a local optimum.A modification of Newton’s method for solving nonlinear equations with a singular Jacobian matrix is introduced.We used QMn-M (a modification of Newton’s method from Lv et al. [15]) combined with a simplified Newtonian iteration (SNR).We used the length error threshold to reduce the frequency of ANN calls to improve the efficiency of real-time control.

First, ABC–BPNN [16] was employed to achieve outstanding prediction values, adjust them to the requirements of this hybrid strategy, and improve the overall strategy convergence and computing performance. Second, the QMn-M algorithm [15] can effectively solve the problem of a numerical method for solving forward kinematics suffering from a singular Jacobian matrix. Third, the global Newton–Raphson method with monotonic descent (GNRDM) proposed by Yang et al. [17] was combined with SNR to improve the efficiency of the algorithm [12]. Inspired by this technique, we used QMn-M combined with SNR to ensure the accuracy of numerical solutions. Moreover, it can reduce the computation time and iterations in overall algorithm. Lastly, the length error threshold $\varepsilon_{0}$ was designed into the overall algorithm process to determine whether the previous set of solutions $Q_{pre}$ satisfied the required value of the current iteration. If this is the case, it is directly used to carry out SNR. As a result, the frequent invocation of ANNs can be reduced. The overall operation efficiency and the problem-solving efficiency of ANNs in real-time motion control can be improved.

The structure of this paper is as follows: In Section 2, we describe the establishment of the kinematics model of GSP, including inverse kinematics problem analysis and the established FKP equations of GSP. The structure of ANNs and the proposed ABC–BPNN applied on the FKP is described in Section 3. Section 4 mainly describes the design and analysis of numerical algorithms. Section 5 outlines a comparative experiment that was conducted on a neural network and numerical algorithm. Lastly, Section 6 concludes the paper.

## 2. Kinematics Model of GSPs

### 2.1. Inverse Kinematics

Compared with the FKP, the inverse kinematics problem (IKP) is relatively easy. The GSP consists of two platforms that are connected by several links, as shown in Figure 1. The base platform is fixed to the ground, and the mobile platform works as the end effector. The motion of six links causes the movement of the end effector.

According to Figure 1, the coordinates of the six lower hinge points are defined as $A_{i}\left(i=1,\;2,\ldots \ldots ,6\right)$, the coordinates of the six upper hinge points are $B_{i}\left(i=1,\;2,\ldots \ldots ,6\right)$. $L_{i}\left(i=1,\;2,\ldots \ldots ,6\right)$ is the length of the link between the connection point $A_{i}$ and $B_{i}$ at initial state. The radii of the six lower- and higher-hinge points are $R_{1}$ and $R_{2}$ respectively, as shown in Figure 2.

The overall structure of 6–6 GSP and its initial state coordinate system are shown in Figure 2. For the base coordinate system $\left\{A\right\}$, defined by its base platform and its origin $O_{A}$ in the platform’s geometric center. The moving coordinate system $\left\{B\right\}$ is attached to the mobile platform, with its origin $O_{B}$ at the platform’s geometric center.

The IKP is to calculate the value of joint variable $L_{Oi}$ by giving the pose of the moving coordinate system. The pose of moving coordinate system $\left\{B\right\}$ is expressed as follows: (1)$$Q={\left[\theta^{T}|O_{B}^{T}\right]}^{T}={\left[\alpha ,\beta ,\gamma ,x,y,z\right]}^{T}$$
where x,y,z are the surge, sway, and heave lines of the mobile platform respectively; and α,β,γ are the roll, pitch, and yaw angles of the mobile platform.

Equation (Equation 3), which is the IKP’s formula for the GSP, can be used to solve the displacement of *i*-th actuator using the geometric technique.
(2)$$L_{Oi}={\left[L_{O1},L_{O2},L_{O3},L_{O4},L_{O5},L_{O6}\right]}^{T}$$
(3)$$L_{Oi}=f_{i}\left(Q\right)=\left|\right|{}^{A}R_{B}B_{i}+O_{B}-A_{i}\left|\right|-L_{li}$$
where $L_{Oi}$ denotes the displacement of the *i*-th link, $L_{li}$ indicates the length of *i*-th link when *Q* is the initial state, and *R* is the rotation matrix in the (Pitch-Roll-Yaw) Euler angle representation, as follows: (4)$$\begin{array}{cc} R_{PRY}\left(\alpha ,\beta ,\gamma \right) & =\left[\begin{array}{ccc} c\gamma & -s\gamma & 0\\ s\gamma & c\gamma & 0\\ 0 & 0 & 1 \end{array}\right]\left[\begin{array}{ccc} c\beta & 0 & s\beta \\ 0 & 1 & 0\\ -s\beta & 0 & c\beta \end{array}\right]\left[\begin{array}{ccc} 1 & 0 & 0\\ 0 & c\alpha & -s\alpha \\ 0 & s\alpha & c\alpha \end{array}\right]\\ \; & =\left[\begin{array}{ccc} c\beta c\gamma & s\alpha s\beta c\gamma -c\alpha s\gamma & c\alpha s\beta c\gamma +s\alpha s\gamma \\ c\beta s\gamma & s\alpha s\beta s\gamma +c\alpha s\gamma & c\alpha s\beta s\gamma -s\alpha c\gamma \\ -s\beta & s\alpha c\beta & c\alpha c\beta \end{array}\right] \end{array}$$
where cα is cosα, sα is sinα, etc.

### 2.2. FKP Equations

The numerical algorithm’s solution necessitates the construction of a set of nonlinear equations for iteration and root finding. To establish the FKP equations of GSP, IKP’s formula of GSP, Equation (Equation 3) is needed. $f_{i}\left(Q\right)$ is the function mapping the pose from Cartesian space to the displacement of the *i*-th actuator $L_{Oi}$ in joint space. The FKP of GSP is to solve corresponding pose *Q* by giving $L_{Oi}$; we can construct Equation (Equation 5) to form a set of nonlinear equations whose constructed equations are Equation (Equation 6).
(5)$$F_{i}\left(Q\right)=f_{i}\left(Q\right)-L_{ni}=0$$
(6)$$\left\{\begin{array}{c} f_{1}\left(Q\right)-L_{n1}=0\\ f_{2}\left(Q\right)-L_{n2}=0\\ f_{3}\left(Q\right)-L_{n3}=0\\ f_{4}\left(Q\right)-L_{n4}=0\\ f_{5}\left(Q\right)-L_{n5}=0\\ f_{6}\left(Q\right)-L_{n6}=0 \end{array}\right.$$
where $L_{ni}$ is the input value of the FKP (the displacement of the *i*-th actuator).

## 3. Artificial Bee Colony-Based BP Neural Network Algorithm

Artificial neural networks (ANNs) have good nonlinear mapping ability, and can realize the mapping relationship between the displacement of several links and the end-effector pose without considering the intermediate operation process. In the ANN-based IKP method, data were obtained by IKP for sample training. $L_{Oi}$ was the input, *Q* was the output, and only approximate output values could be obtained. The backpropagation neural network (BPNN) is a widely used artificial neural network in GSPs [9].

### ABC–BP Neural Network

According to the mapping relation of $f\left\{L_{O1},L_{O2},L_{O3},L_{O4},L_{O5},L_{O6}\right\}\to \left\{\alpha ,\beta ,\gamma ,x,y,z\right\}$, a BPNN with 6 input nodes, 6 output nodes, and 1 hidden layer was designed as shown in Figure 3. Sigmoid function was chosen to be the activation function.

The BPNN model achieved promising performance and flexible network structure. However, it had several deficiencies that prevented it from completing this mission. To optimize the BPNN, the artificial bee colony (ABC) algorithm was used to ensure the stability of the deviation between the predicted and real values of the IKP’s dataset of BPNN training, and to enhance the model training effect, namely, ABC–BPNN [18,19,20,21].

For the BPNN model, the initialization weights and thresholds are given as follows, respectively:$$W_{1}=\left(\begin{array}{cccccc} 0.0662 & -1.6832 & -1.527 & -1.4254 & -1.0288 & 2.0796\\ -0.9240 & 2.5984 & -1.9986 & -1.1251 & 1.9704 & -0.8427\\ -2.9220 & 2.1942 & -0.7272 & 0.4587 & -1.2002 & -2.6277\\ -1.2651 & -0.7931 & 1.6843 & 1.4514 & -2.1420 & -2.4401\\ 2.2850 & -2.6858 & 1.3005 & 1.4514 & 2.6247 & 1.6386\\ 2.6616 & 1.1087 & 0.6891 & -1.3788 & 0.1340 & 2.3670\\ 2.6722 & -1.5912 & 1.3156 & -2.1011 & 1.9623 & 0.4815\\ 1.9344 & -1.4598 & 1.9829 & 2.7022 & -1.8650 & 1.4510\\ 1.9723 & -1.9167 & 0.1324 & -2.7276 & 0.9876 & 0.6103\\ 2.5931 & -2.5397 & 0.0078 & -0.7973 & 1.4428 & 1.8874 \end{array}\right)B_{1}=\left(\begin{array}{c} 0.9841\\ 0.7650\\ -2.2820\\ -2.1350\\ -1.4340\\ 1.5495\\ 0.3450\\ 2.5649\\ 2.2165\\ -0.6414 \end{array}\right)$$
$$W_{2}=\left(\begin{array}{cccccc} -1.7583 & -1.4096 & 2.8489 & 0.9987 & -1.8899 & -0.0096\\ 1.8149 & -0.7841 & 1.9333 & 0.0338 & 2.6185 & -1.6147\\ 1.4687 & -1.8757 & 2.7148 & 0.8056 & 0.9578 & 1.2417\\ 2.6779 & -0.0114 & -1.9383 & 2.7488 & 1.1466 & 0.4589\\ 0.1704 & -1.2281 & 0.6174 & 0.5256 & -0.8983 & 1.1637\\ 0.6082 & -0.7252 & -0.5100 & 2.5182 & -1.6204 & -1.5583\\ 1.1490 & 1.1365 & 2.4286 & 0.9827 & 0.9567 & -1.1117\\ 1.9906 & -0.0194 & -2.0536 & 2.4177 & 2.2674 & -0.1703\\ 1.1783 & 2.2060 & -1.9097 & 1.8553 & -0.2285 & -0.8113\\ -0.3779 & 0.7793 & -1.4021 & 2.8468 & 0.5574 & 1.5143 \end{array}\right)B_{2}=\left(\begin{array}{c} 1.4588\\ -0.9130\\ 1.0489\\ 0.3667\\ 2.6758\\ -1.6079 \end{array}\right)$$

Due to its benefits of fast convergence and global search, the ABC method is frequently used to solve optimization issues. Bees and food sources are the two most important components of the ABC algorithm. Scout, employed, and onlookers are the three sorts of bees. Scout bees are entrusted to search for food sources at random, while employed and onlooker bees are in charge of nectar mining [22,23]. Numerous studies showed that the ABC algorithm outperforms the genetic algorithm (GA), particle swarm optimization (PSO), and other methods in terms of accuracy and convergence rate [24,25]. In fact, the principle of the ABC algorithm is dealing with the problem of function optimization by simulating the nectar-gathering mechanism of actual bees. The ABC algorithm iterates continuously, compares the advantages and disadvantages of the problem, keeps the good individuals and eliminates the bad ones, and constantly approaches the global optimal solution. The ABC algorithm replaces the solution to the optimal honey of the colony with the solution to the optimal weights and thresholds of the neural network. The optimized BPNN is trained by the global optimal weights and thresholds provided by ABC. The whole ABC–BPNN algorithm is shown in Algorithm A1. The specific steps are as follows:Build a BPNN model.Initialize the parameters of the ABC algorithm: self-variable dimension *D*, population number *N* (the number of employed bees is $N_{1}$, the number of onlookers is $N_{2}$, and the number of solutions is $N_{3}$), maximal cycle number *G*, and threshold of iteration number for scout bees limit.
(7)$$D=N_{input}\times N_{hidden}+N_{hidden}+N_{hidden}\times N_{output}+N_{output}$$
(8)$$N=N_{1}+N_{2}=2N_{3}$$
where $N_{input}$ is the number of nodes in the input layer, $N_{hidden}$ is the number of nodes in the hidden layer, and $N_{output}$ is the number of nodes in the output layer.Employed-bee search. New solution $V_{i}\left(j\right)$ is generated by employed bees searching the field and is compared with old solution $X_{i}\left(j\right)$ by using the principle of the greedy algorithm. The strategy compares the size by calculating the fitness value. If the fitness value of the new solution is greater than that of the solution, the solution is replaced by the updated solution. Otherwise, the number of updates is failures + 1. The fitness value is calculated with Equation (Equation 10), and $V_{i}\left(j\right)$ is calculated with Equation (Equation 9).
(9)$$V_{i}\left(j\right)=X_{i}\left(j\right)+\left(-1+2rand\left(0,1\right)\right)\times \left(X_{i}\left(j\right)-X_{r1}\left(j\right)\right)$$
where $X_{i}\left(j\right)$ is a random selection of other employed bees, different from themselves, $r_{1}\in \left[1,2,3,\ldots ,N_{3}\right]$.
(10)$$f\left(x_{i}\right)=\left\{\begin{array}{cc} \frac{1}{\left(MSE_{i}+1\right)},\;\; & MSE_{i}>0\\ 1\;\;\;\;\;\;\;,\;\; & MSE_{i}<0 \end{array}\right.$$
(11)$${MSE}_{i}=\sum_{m=1}^{n}\frac{{\left(y_{im}-{\hat{y}}_{im}\right)}^{2}}{n}$$
where ${MSE}_{i}$ is the mean square error generated by training the *i*-th solution in the BPNN, ${\hat{y}}_{im}$ indicates the predicted value during training in the BPNN, $y_{im}$ indicates the actual value, and *n* indicates the total number of training samples.
(12)$$X_{i}^{t+1}=\left\{\begin{array}{cc} V\;,\;\; & f\left(V\right)<f\left(X_{i}^{t}\right)\\ X_{i}^{t},\;\; & f\left(V\right)\geq f\left(X_{i}^{t}\right) \end{array}\right.$$
where $X_{i}^{t}$ represents the *i*-th individual in the population of the *t* generation.Onlookers search. Onlookers calculate possible value $P_{i}$ with Equation (Equation 13), and use the roulette to find a new solution among existing ones.
(13)$$P_{i}=\frac{f\left(x_{i}\right)}{\overset{N_{3}}{\underset{n=1}{\sum }}f\left(x_{n}\right)}$$Scout-bee search. The rapid convergence of employed and scout bees may lead to a decrease in the overall diversity of the population. In order to avoid the population entering the local optimum, the search mechanism of scout bees was designed on the basis of the ABC algorithm. If the number of update failures exceeds limit, the solution is discarded and replaced by a new solution generated by Equation (Equation 14), and the number of update failures is initialized to zero. In this paper, the value of limit is the product of $N_{3}$ and *D*.
(14)$$X_{i}=X_{min}+rand\left(0,1\right)\left(X_{max}-X_{min}\right)$$If the current cycle number is greater than maximal cycle number *G*, Step 3 is repeated. Otherwise, the solution with the maximal fitness is output at the end of the training.According to the optimal solution provided by the ABC algorithm, the initial weights and thresholds of the BPNN are obtained. Subsequently, the ABC–BPNN model is trained and tested with the sample data to achieve the pose prediction of GSP.

## 4. Analysis and Design of the Numerical Algorithm

### 4.1. Newton–Raphson Algorithm

In the traditional hybrid algorithm (BPNN and numerical algorithm), Newton–Raphson is used as the main method for numerical solutions. Some other improved Newtonian methods are used for the numerical iteration of optimization algorithms, as shown in Figure 4.

Newton–Raphson method:The formula of the NR method for solving nonlinear equations of FKP is:
(15)$$Q_{n+1}=Q_{n}-J^{-1}Q_{n}F\left(Q_{n}\right)\;\;\;\;\;\;\left(n=0,1,2,3\ldots \ldots \right)$$
where *J* is a Jacobian matrix, and its formula is:
(16)$$J=\left[\begin{array}{ccc} \frac{\partial F_{1}}{\partial \alpha_{1}} & \ldots & \frac{\partial F_{1}}{\partial z_{1}}\\ \vdots & \ddots & \vdots \\ \frac{\partial F_{6}}{\partial \alpha_{6}} & \ldots & \frac{\partial F_{6}}{\partial z_{6}} \end{array}\right]$$Newton downhill method:The formula for the Newton downhill (NR-dh) method for solving nonlinear equations of FKP is:
(17)$$Q_{n+1}=Q_{n}-\lambda J^{-1}Q_{n}F(Q_{n})\;\;\;\;\;\;\left(0<\lambda \leq 1\right)$$
where λ is the descent factor; when λ=1, it is the NR method.When $||F\left(Q_{n}+1\right)||<||F\left(Q_{n}\right)\left|\right|$, λ remains the same. On the other hand, $\lambda =2^{-t},\left(t=0,1,2\ldots \ldots \right)$.

### 4.2. Improved Newton’s Method for Nonlinear Singular Equations

Neither the traditional nor the above method can solve the singular Jacobian matrix of FKP nonlinear equations. According to Equation (Equation 15), a new set of *J* is formed continuously in the process of the solving iteration, and its inverse is required. In the process of iteration, when *J* forms a singular matrix, $Rank\left(J\right)<6$, its inverse cannot be solved. Therefore, the general solution formula of NR method should be: (18)$$Q_{n+1}=Q_{n}-J^{+}Q_{n}F(Q_{n})\;\;\;\;\;\;\left(n=0,1,2,3\ldots \ldots \right)$$
where continuous iterating pseudoinverse matrix $J^{+}$ slows down the solution process, affects the overall efficiency, and cannot guarantee accuracy of computation. To solve these problems, any value that is input into the feasible motion space of the mechanism can have a solution. Therefore, an improved Newton’s method for solving nonlinear singular equations was introduced [15]. Lv et al. [15] combined the methods that were proposed in Kou et al. [26], and Howk et al. [27]; a diagonal matrix containing parameters was added to the singular Jacobian matrix to change the irreversible characteristics of the Jacobian matrix in the iterative process, and the convergence order of this new singular algorithm was second. The equation proposed in Kou et al. [26] is as follows: (19)$$Q_{n+1}=Q_{n}-{\left[diag\left({\mu_{i}}^{\left(n\right)}F\left(Q_{n}\right)\right)+J\right]}^{-1}F\left(Q_{n}\right)\;\;$$
where ${\mu_{i}}^{\left(n\right)}\in R\setminus \left\{0\right\}$. when *J* is for the singular matrix, the $diag\left({\mu_{i}}^{\left(n\right)}F\left(Q_{n}\right)\right)$ join can solve this problem.

The formula of the improved Newton’s method for nonlinear singular equations is: when n=0, the formula is: (20)$$Q_{0}^{*}=Q_{0},\;Q_{1}=Q_{0}-{\left[diag\left({\mu_{i}}^{\left(0\right)}F(Q_{0})\right)+F^{\prime }\left(\gamma \left(Q_{0}\right)+\left(1-\gamma \right){Q_{0}}^{*}\right)\right]}^{-1}F(Q_{0})$$
when n≥1, the formula is: (21)$$\begin{array}{cc} \; & Q_{n}^{*}=Q_{n}-{\left[diag\left({\mu_{i}}^{\left(n\right)}F\left(Q_{n}\right)\right)+F^{\prime }\left(\gamma Q_{n-1}+\left(1-\gamma \right)\right)Q_{n-1}^{*}\right]}^{-1}F\left(Q_{n}\right),\\ \; & Q_{n+1}=Q_{n}-{\left[diag\left({\lambda_{i}}^{\left(n\right)}F\left(Q_{n}\right)\right)+F^{\prime }\left(\gamma Q_{n}+\left(1-\gamma \right)\right)Q_{n}^{*}\right]}^{-1}F\left(Q_{n}\right) \end{array}$$
where, when γ=12, it is the QMn-M algorithm. When γ=0, it is the PC-M algorithm. When γ=1, the two-step iterative Equations (20) and (21) change into the single-step iterative Equation (Equation 19), which is Newton’s improvement on the standard.

The recommended algorithm is the QMn-M algorithm at γ=12, which performed better on the FKP of GSP, with better iterations and total running time than those of the others. A comparison of several algorithms is shown in Table 1.

### 4.3. Simplified Newton Iteration to Optimize QMn-M Algorithm

The QMn-M algorithm solves singular Jacobian matrix *J* and reduces its problem-solving speed. To ensure the global performance, and its operation speed and accuracy, we used the simplified Newton iteration (SNR) method to optimize the QMn-M algorithm.

Simplified Newton Iteration: (22)$$Q_{n+1}=Q_{n}-CF\left(Q_{n}\right),\;\;\;\;\;\;C=J^{-1}\left(Q_{\varepsilon }\right)$$
where *C* is a constant Jacobian matrix. When it is not satisfied: (23)$$||F\left(Q_{n+1}\right)||<||F\left(Q_{n}\right)\left|\right|$$
to renew the Jacobian matrix.

The QMn-M and SNR algorithm can effectively improve the time efficiency of the algorithm and guarantee the accuracy of the value, as shown in Figure 5. Compared with the traditional hybrid algorithm, it can solve the global pose problem.

### 4.4. FKP of Real-Time Control Improvement

In the control of a 6-DOF parallel robot, a real-time solution of FKP is more practical. The control cycle and sampling interval are usually milliseconds. Therefore, the solution of FKP connected between control cycles is approximate. According to the deviation-driven algorithm on real-time FKP [12], and by Equation (24), $Q_{pre}$ (the value of the last solution) is reused, and ABC–BPNN calls are reduced. If the formula is satisfied, $Q_{pre}$ is used to directly carry out a simplified Newton iteration. Otherwise, ABC–BPNN is called to give an approximation.
(24)$$\Delta L_{max}=max\left|f_{i}\left(Q_{pre}\right)-L_{ni}\right|$$
(25)$$\Delta L_{max}<\varepsilon_{0}$$
where $\varepsilon_{0}$ is the length error threshold that controls whether to call ANNs. The whole hybrid algorithm is shown in Figure 6 and Algorithm A2.

## 5. Experiment

The experimental process can be divided into three parts:Part 1: Inverse kinematics solution. The corresponding data are obtained to form the dataset and normalized.Part 2: Training for neural networks. Test samples are input into the trained model for pose prediction.Part 3: Numerical iteration. Predicted results are input as the numerical algorithm’s initial point to acquire the precise output.

Several experiments were conducted on the GSP (6-UCU) as shown in Figure 1. The positions of hinge points are given in Table 2. The maximal iterations $N_{max}$ were set to 20, and the permissible error was $10^{-4}$ mm. The following experiments were carried out in the environment of MATLAB R2019b, and the operating system was Windows 10. When $Q={\left[0,0,0,0,0,0\right]}^{T}$, the initial value of each links variable was:
$L_{i}={\left[1306.7130,1306.7130,1306.7130,1306.7130,1306.7130,1306.7130\right]}^{T}$; the unit is millimeters.

### 5.1. Data Acquisition

A specific number of poses inside the workspace should be chosen before the training process, and the associated actuator displacements should be acquired by solving IKP with the corresponding actuator displacement Equation (Equation 3). The sample sets for training ANNs are formed by combining the two groups of data and then reversing their mappings. Essentially, ANNs use calculated data in a joint space as inputs, and corresponding pose data in Cartesian space as predicted outputs.

### 5.2. Neural Network Experiment

To train the ANNs, the network/data management toolbox in MATLAB R2019b was used. In the neural network comparison experiment, 300 groups of continuous data were collected from the machine. We divided the dataset at a ratio of 7:3. The training set consisted of 210 groups, and the test set consisted of 90 groups. We introduced a genetic algorithm–backpropagation neural network (GA-BPNN) to compare with our ABC–BPNN and the traditional BPNN. The BPNN, GA–BPNN, and ABC–BPNN parameter settings were the same, where the learning rate was 0.001, and set epochs were 200. In this experiment, the three neural network models were compared through metrics, as shown in Table 3. The average error of each variable predicted by the neural network model is given in Table 4. In this paper, we calculated the error through the numerical deviation of the corresponding data. The error calculation equation used is as follows: (26)$$Error=|\hat{Q_{i}}-Q_{i}|,\;\;\;\left(i=1,2,3,\ldots ,6\right)$$
(27)$$AverageError=\frac{1}{n}\sum_{m=1}^{n}\left|{\hat{Q}}_{mi}-Q_{mi}\right|,\;\;\;\left(i=1,2,3,\ldots ,6\right)$$
where $\hat{Q_{i}}$ represents the predicted value of a pose variable, $Q_{i}$ represents the real value of a pose variable, and *m* represents the individual number of the same variable in ${\hat{Q}}_{mi}$.

Figure 7 shows the predicted values of the three models in a test set. A group of data were selected for comparison, and the pose was the predicted value of the neural network. The experimental results in the figure demonstrate that ABC–BPNN had a better prediction ability than that of the traditional BPNN or GA–BPNN. Figure 8 compares the error of the surge(x) predicted value of the neural network models with 10 groups of sample data. the error calculation of the surge variable was performed with Equation (Equation 26). Through the optimized neural network, its error value obviously decreased, as shown in the figure. By using a heuristic algorithm, the network model’s prediction performance was enhanced, and ABC–BPNN was more able to adjust to changes in the process.

According to the metrics given in Table 3, the ABC algorithm optimized the initialization weights well, which improved the prediction ability of the BPNN model. Compared with GA–BPNN and BPNN, the RMSE index of ABC–BPNN decreased by 42.1% and 31.4%, respectively, and the $R^{2}$ performance of ABC–BPNN improved by 11.0% and 5.6%, respectively. The average error of the three models is given in Table 4; the maximal improvement of ABC–BPNN error optimization was 46.3%. By comparing the average error, the ABC algorithm could reduce the average error of various variables, and its effect was better than that of the genetic algorithm. Experimental results show that ABC–BPNN could reduce the dependence on the dataset and the prediction error of the neural network.

### 5.3. Numerical Algorithm Experiment

We used the data obtained by the neural network to conduct experiments in MATLAB R2019b. On a PC with a 2.30 GHz processor and 8 GB RAM, the FKP that corresponded to the obtained data was solved. An error threshold $\varepsilon_{2}$ was used to control the QMn-M algorithm in the experiment, and an error threshold $\varepsilon_{1}$ was subsequently used to control the SNR algorithm. The number of iterations and overall operation time fluctuate depending on $\varepsilon_{1}$. The design of the error threshold can be changed with specific data until the value that renders it the most efficient is found. Error threshold $\varepsilon_{1}$ is used to govern the end of the SNR algorithm and the entire process. Its value is generally accurate to four decimal places (as shown in Table 5, The precise level was $10^{-4}$).

After the neural network experiment, the predicted values were iterated numerically. In this section, the average numbers of iterations, required operation time, and different iterative methods are compared. Table 5 shows the comparison of seven iteration methods, giving the average number of iterations and the required time. As shown in Table 5, the required time and average number of iterations of the proposed QMn-M and SNR were reduced, and its performance was better than that of the same second-order NR algorithm after being optimized by SNR. In Table 6, the traditional NR, and NR-dh with QMn-M and PC-M are compared. QMn-M and PC-M could solve nonlinear singular equations, and the advantages of this algorithm in solving the forward kinematics problem are demonstrated. Experiments showed the feasibility of QMn-M and PC-M in solving singular matrix data. (${\mu_{i}}^{\left(n\right)},{\lambda_{i}}^{\left(n\right)}$) about $Q_{n}^{*}$ and $Q_{n+1}$ in Equation (Equation 21) were set to be the same parameters. In the QMn-M experiment, different (${\mu_{i}}^{\left(n\right)},{\lambda_{i}}^{\left(n\right)}$) values caused different levels of performance, and unsuitable (${\mu_{i}}^{\left(n\right)},{\lambda_{i}}^{\left(n\right)}$) values may render it divergent and unsolvable. Figure 9 shows the comparative experiment of the operation time required by four algorithms in 10 groups of data. As described in the Figure 9, QMn-M and Newton–Raphson had the best performance, followed by PC-M, and NR-dh had the worst performance. Figure 10 is a comparison of QMn-M, QMn-M and SNR, and the traditional NR method. It shows that SNR combined with a numerical algorithm could greatly improve the efficiency of the algorithm and better guarantee accuracy.

Furthermore, we found the percentage improvement performance of other iterative algorithms by comparing with QMn-M and SNR, which is illustrated in Table 7. Compared to NR-hd, the maximal percentage improvement in performance for the average number of iterations and required time was 68.8% and 70.60%, respectively. Compared to NR, the minimal percentage improvement in performance for the average number of iterations and required time was 14.4% and 13.9%, respectively.

## 6. Conclusions

This paper mainly optimized the traditional hybrid algorithm for FKP. The ABC–BPNN model was used to train the IKP’s value to obtain a good initial value, and the QMn-M and SNR algorithm was used to iteratively calculate the excellent initial value. The comparison chart in the numerical algorithm experiment shows that the QMn-M and SNR algorithm was superior to other methods in terms of iterations and operation time. It also has the ability to calculate a singular matrix. For continuous samples, its solution does not need to continuously call ABC–BPNN, which greatly reduces the overall operation time required, and makes it convenient to solve on the real-time platform. This new hybrid algorithm can not only ensure that the GSP can have solutions in feasible space, but also meet the speed and accuracy of real-time computation. This algorithm also has portability, which is beneficial to solve the FKP for different structural platforms.

## Figures and Tables

**Figure 1 sensors-22-05318-f001:**
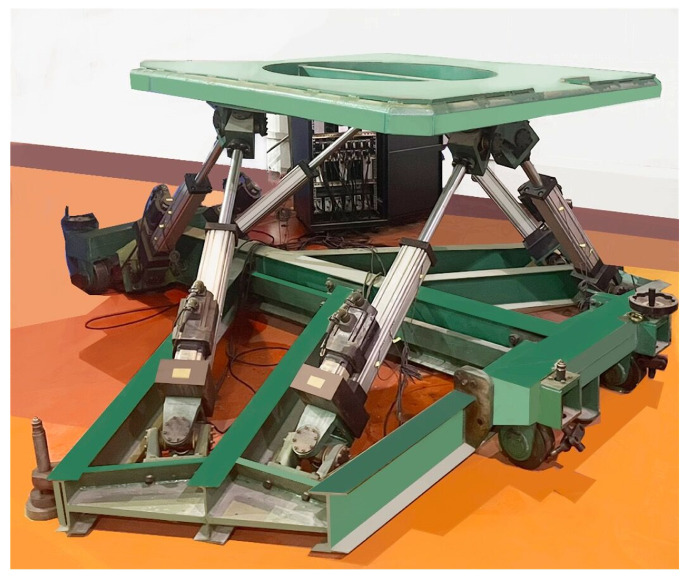
Example of a 6-UCU GSP.

**Figure 2 sensors-22-05318-f002:**
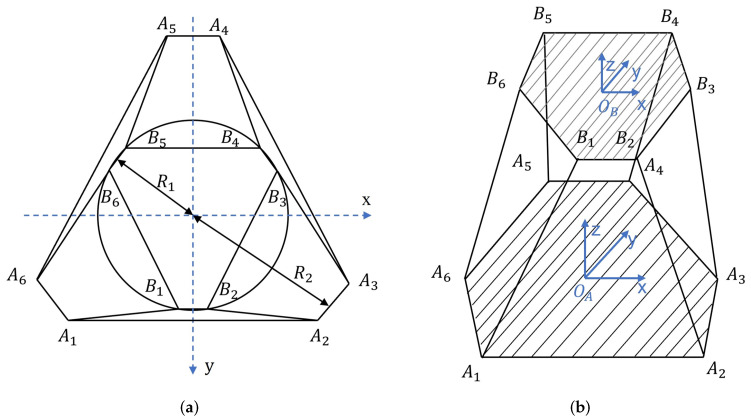
Schematic of 6-UCU GSP. (**a**) Schematic of the universal joint positions; (**b**) 6–6 GSP model and coordinate systems.

**Figure 3 sensors-22-05318-f003:**
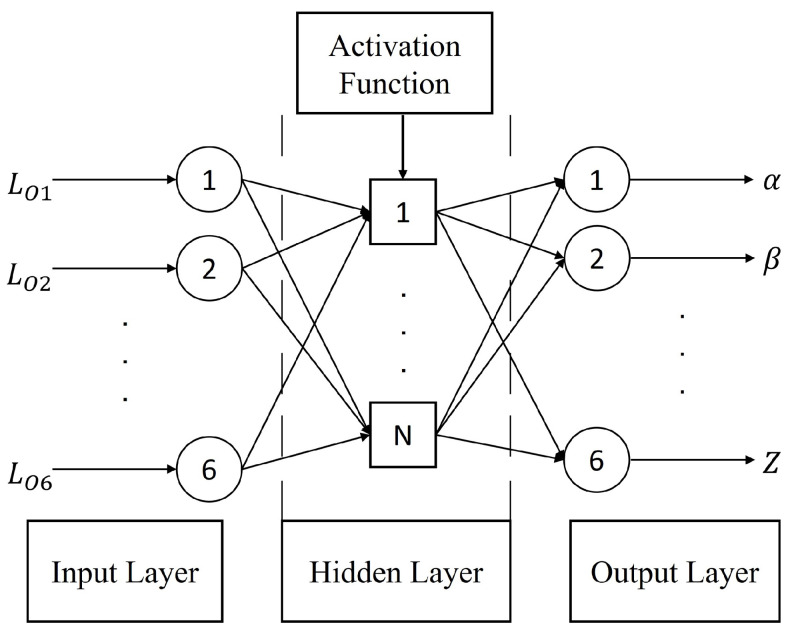
Structure of the BP neural network.

**Figure 4 sensors-22-05318-f004:**
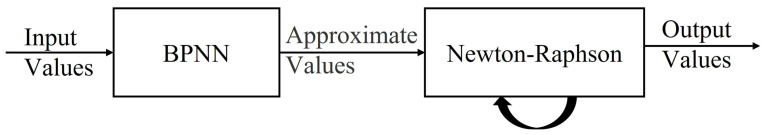
Traditional hybrid algorithm.

**Figure 5 sensors-22-05318-f005:**
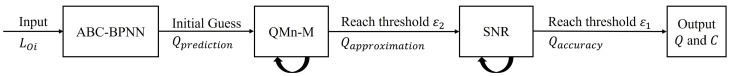
QMn-M and SNR algorithm.

**Figure 6 sensors-22-05318-f006:**
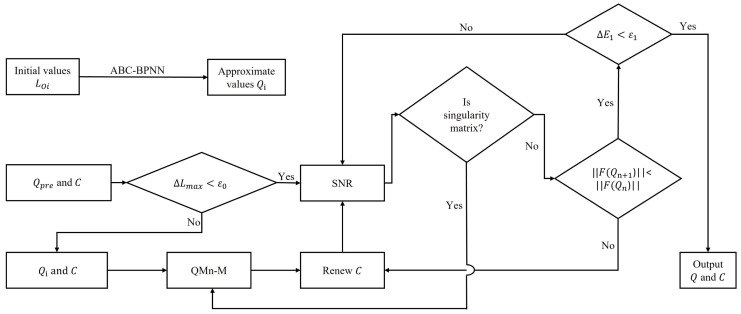
Algorithm for forward kinematics problem of 6-DOF. $\Delta E_{1}$ refers to the difference between $Q_{n+1}$ and $Q_{n}$ of each iteration, and $\varepsilon_{1}$ is the threshold for controlling the end of the iteration.

**Figure 7 sensors-22-05318-f007:**
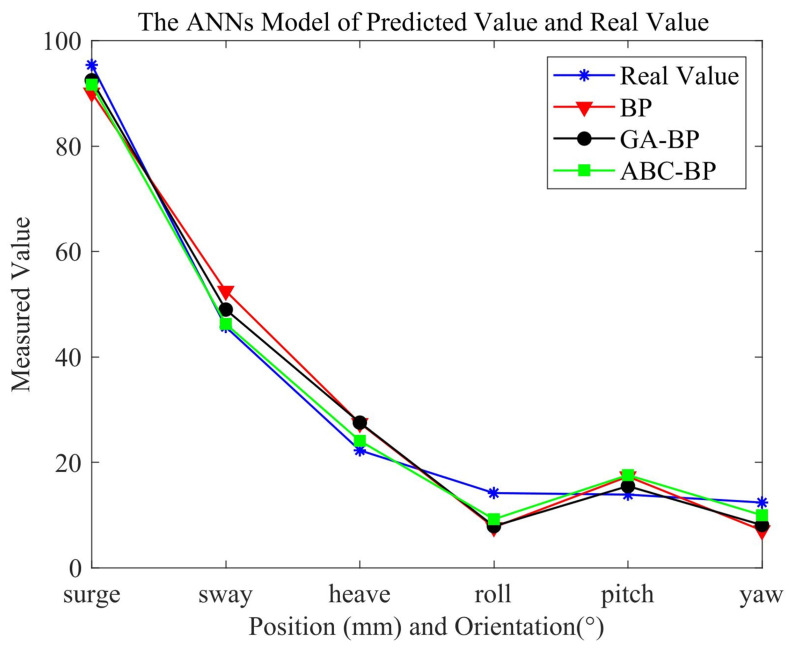
Predicted and real values of ANN model.

**Figure 8 sensors-22-05318-f008:**
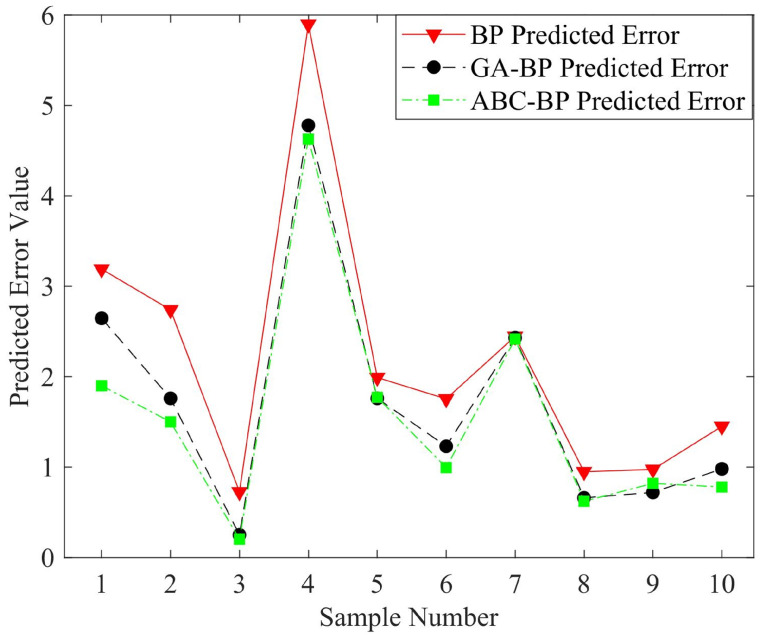
Predicted error value of surge(x).

**Figure 9 sensors-22-05318-f009:**
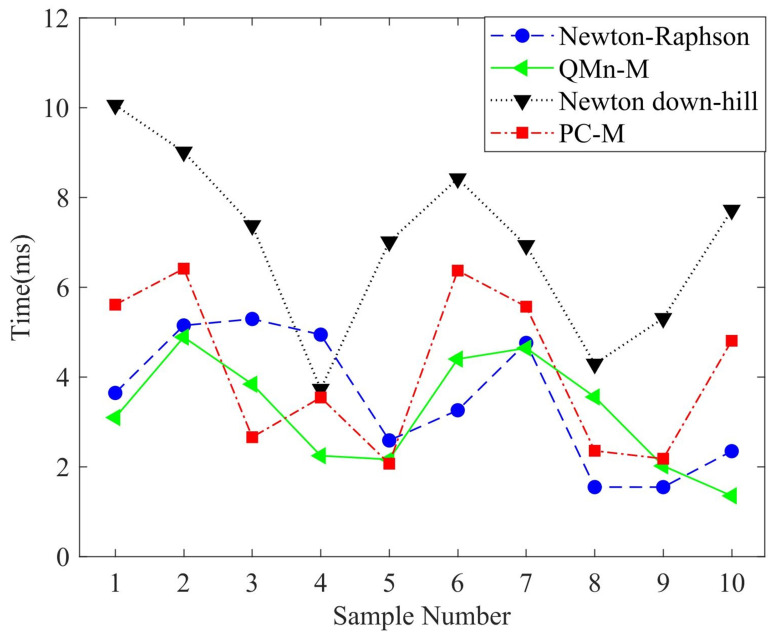
Comparison of algorithm operation time.

**Figure 10 sensors-22-05318-f010:**
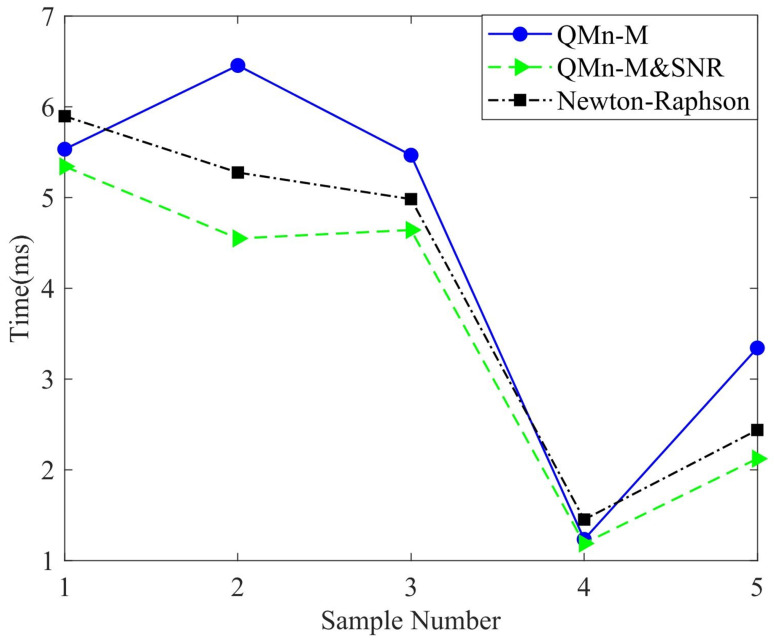
Comparison of algorithm operation time for QMn-M and SNR.

**Table 1 sensors-22-05318-t001:** Analysis of the advantages and disadvantages of several numerical algorithms.

Numerical Algorithms	Advantages	Disadvantages
Newton–Raphson	Easy to implement and fast convergence.	Sensitive to initial value, and the iteration is easy to diverge.
Newton downhill	Reduces sensitivity to initial values.	The strong coupling of the Jacobian matrix reduces the iteration speed in FKP equations.
QMn-M or PC-M	It can solve the nonlinear equations of a singular Jacobian matrix, and only two function values need to be calculated in each iteration step ^1^.	The amount of matrix calculation is increased, and more parameter settings are added.

^1^ With the QMn-M or PC-M algorithm, only two function values need to be calculated in each iteration step because, in Equation (21),
$F^{\prime }(\gamma Q_{n-1}+(1-\gamma )Q_{n-1}^{*})$ in the last iteration formula is in Equation (21). The only need is to calculate two functions: $F^{\prime }(\gamma Q_{n-1}+(1-\gamma )Q_{n-1}^{*})$.

**Table 2 sensors-22-05318-t002:** Position of hinge points.

		Hinge Point Numbers ^1^ (mm)				
	1	2	3	4	5	6
$X_{A}$	−250.0	250.0	1350.0	1100.0	−1100.0	−1350.0
$Y_{A}$	1414.5	1414.5	−490.7	−923.7	923.7	−490.7
$Z_{A}$	0.0	0.0	0.0	0.0	0.0	0.0
$X_{B}$	−525.0	525.0	625.0	100.0	−100.0	−625.0
$Y_{B}$	418.5	418.5	245.3	−663.9	−663.9	245.3
$Z_{B}$	0.0	0.0	0.0	0.0	0.0	0.0

^1^ (*X*_*A*_, *Y*_*A*_, *Z*_*A*_) is the value of base coordinate system {*A*}. (*X*_*B*_, *Y*_*B*_, *Z*_*B*_) is the value of moving coordinate system {*B*}.

**Table 3 sensors-22-05318-t003:** Comparison of four metrics.

	Model Name		
	**BP**	**GA-BP**	**ABC-BP**
$Train\_R^{2}$	0.7997	0.8494	0.8972
$Test\_R^{2}$	0.7961	0.8377	0.8945
Train_MAE	4.3541	3.3694	2.8674
Test_MAE	4.5543	3.4001	2.8784
Train_RMSE	5.7324	4.8546	3.3291
Test_RMSE	5.8024	4.8916	3.3548
Train_MAPE	11.8970%	9.5493%	7.9761%
Test_MAPE	12.0483%	9.6759%	7.9801%

**Table 4 sensors-22-05318-t004:** Average error of neural network model.

	Average Error of BP	Average Error of GA–BP	Average Error of ABC–BP
**Surge (mm)**	4.25	3.15	2.88
**Sway (mm)**	4.54	3.45	2.63
**Heave (mm)**	4.68	3.67	2.51
**Roll (°)**	3.49	2.96	2.77
**Pitch (°)**	3.55	2.94	2.93
**Yaw (°)**	3.58	3.01	2.84

**Table 5 sensors-22-05318-t005:** Performance comparison of methods.

	IN ^1^/IN1 ^2^	Time (ms)
**Newton–Raphson**	3.345/0	3.069
**Newton downhill**	9.177/0	8.997
**QMn-M**	3.447/0	3.151
**PC-M**	4.015/0	3.934
**NR-dh and SNR**	5.354/3.173	4.522
**QMn-M and SNR**	2.862/1.059	2.641
**PC-M and SNR**	3.409/1.137	3.218

^1^ IN represents the total number of iterations. ^2^ IN1 represents the number of simplified Newton iterations.

**Table 6 sensors-22-05318-t006:** Comparison of numerical algorithms to solve a singular matrix.

	${\mu_{i}}^{\left(n\right)},{\lambda_{i}}^{\left(n\right)}$	Input	Output
**Newton–Raphson**	(NULL, NULL)	${\left(102.9,51.5,\ldots ,16.6\right)}^{T}$	Failure
**Newton downhill**	(NULL, NULL)	${\left(102.9,51.5,\ldots ,16.6\right)}^{T}$	Failure
**QMn-M**	(0.1,0.1)	${\left(102.9,51.5,\ldots ,16.6\right)}^{T}$	${\left(101.7,50.3,\ldots ,15.4\right)}^{T}$
**PC-M**	(0.1,0.1)	${\left(102.9,51.5,\ldots ,16.6\right)}^{T}$	${\left(101.7,50.3,\ldots ,15.4\right)}^{T}$
**Newton–Raphson**	(NULL, NULL)	${\left(97.3,47.2,\ldots ,14.6\right)}^{T}$	Failure
**Newton downhill**	(NULL, NULL)	${\left(97.3,47.2,\ldots ,14.6\right)}^{T}$	Failure
**QMn-M**	(0.1,0.1)	${\left(97.3,47.2,\ldots ,14.6\right)}^{T}$	${\left(95.2,46.7,\ldots ,14.3\right)}^{T}$
**PC-M**	(0.1,0.1)	${\left(97.3,47.2,\ldots ,14.6\right)}^{T}$	${\left(95.2,46.7,\ldots ,14.3\right)}^{T}$
**Newton–Raphson**	(NULL, NULL)	${\left(104.9,50.4,\ldots ,18.1\right)}^{T}$	Failure
**Newton downhill**	(NULL, NULL)	${\left(104.9,50.4,\ldots ,18.1\right)}^{T}$	Failure
**QMn-M**	(0.01,−0.1)	${\left(104.9,50.4,\ldots ,18.1\right)}^{T}$	${\left(103.2,50.1,\ldots ,16.1\right)}^{T}$
**PC-M**	(0.01,−0.1)	${\left(104.9,50.4,\ldots ,18.1\right)}^{T}$	${\left(103.2,50.1,\ldots ,16.1\right)}^{T}$

**Table 7 sensors-22-05318-t007:** Improvement percentages of the proposed method.

QMn-M and SNR	Average Number of Iterations	Required Time
Over Others	% of Improvement	% of Improvement
**Newton–Raphson**	14.4	13.9
**Newton downhill**	68.8	70.6
**QMn-M**	16.9	15.2
**PC-M**	28.7	32.8
**NR-dh and SNR**	46.5	41.5
**PC-M and SNR**	16.0	17.9

## Data Availability

The data presented in this study are available on request from the corresponding author. The data are not publicly available due to privacy.

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
