# Peer review of "A Novel Hybrid Algorithm for the Forward Kinematics Problem of 6 DOF Based on Neural Networks"

_sensors, 2022, doi:10.3390/s22145318_

Round 1
Reviewer 1 Report
This paper analyzed and discussed classical forward kinematics problem of Gough-Steward parallel platforms. An interesting hybrid algorithm based on the combination of a BP neural network optimized by Artificial Bee Colony (ABC) and a modification of Newton’s method was proposed. The ABC was used to provide initial values and the modification of Newton’s method was used to solve the problem when the Jacobian matrix was trapped in singular. The contributions are clear but there are some problems needed to consider for the improvement of the current version.
1. How to select the max interactions in experiments? Please describe the selection foundation detailedly.
2. The 300 groups of continuous data were used in 5.2, how to get these data?
3. In Figure 7, why the predicted precision of roll is the worst compared with other positions and orientations.
4. The control cycle and sampling interval of parallel robot are usually millisecond. While the operation times in Figure 10 and Figure 11 are second. How to use the proposed algorithm in real-time control?
5. The presentation in line 101 “Jacobin matrix cannot solve…” and line 217 “Then, Training after…” should be corrected.
Reviewer 2 Report
The paper entitled "A Novel Hybrid Algorithm for Forward Kinematics Problem of 6-DOF Based on Neural Networks" touches an important topic and results looks promising. However, some fundamental questions arises from the the paper's presentation.
1. However, the author has not provided the parameters for the BPNN in section 3.1.
2. Also the data split of 280 set for train and 20 (6.6%) set for testing. What does authors mean by continuous data? Why the train-test split is so different than academics norm of 70:30 or 60:40?
3. Table 3 consist of test metrics? What about train metrics? It's hard to judge if the model was overfit or underfit in this case. Because there are no training metrics provided.
4. Table 4 are average results or just for one set? In general, average values are provided.
5. Figure 8 the units are not correct for axis. Figure 9 the unit for x axis is position and degree? But where are the values? What is the unit for measurement error? How measurement error (error of surge?) is calculated?
6. How many singular points were compared in Table 5? Ability of a model cannot be proven by comparing just one configuration.
7. "Figure 11, QMn-M & SNR has a perfect speed that makes it superior to others." What does authors mean by perfect speed? In computational field, if the speed of an algorithm has predictable latency, less deviation then it is considered an stable algorithm but statements like 'perfect speed" needs to be justified by supporting example like very low latency or linear complexity (O(n)) etc. In this figure the computation time is correlates with every other algorithm, how is it perfect? Are authors trying to address an real-time computational problem? Please provide the average % improvement in a table format instead of figure when making such claims.
Reviewer 3 Report
1. Results: Recommend to be Major revisions
This paper proposes a new hybrid algorithm based on the combination of an Artificial Bee Colony (ABC) optimized BP neural network (ABC-BPNN) and a numerical algorithm to solve the forward kinematics problem of Gough-Stewart platforms. ABC greatly improves the prediction ability of neural network and can provide a superb initial value to numerical algorithm. In the design of numerical algorithm, a modification of Newton’s method (QMn-M) is introduced to solve the problem that the traditional algorithm model can’t be solved when it is trapped in singular matrix. Results show that the model optimization performance improves by 28.88%, while the MSE index decreases by 46.43%. Experiments show the feasibility of QMn-M in solving singular matrix data and it has a perfect speed that makes it superior to others.
This paper is with none merits for Sensors, i.e., poor writing skills and lacking of insight analysis, it requires some major revisions.
Firstly, for Section 1, authors should provide the comments of the cited papers after introducing each relevant work. What readers require is, by convinced literature review, to understand the clear thinking/consideration why the proposed approach can reach more convinced results. This is the very contribution from authors. In addition, authors also should provide more sufficient critical literature review to indicate the drawbacks of existed approaches, then, well define the main stream of research direction, how did those previous studies perform? Employ which methodologies? Which problem still requires to be solved? Why is the proposed approach suitable to be used to solve the critical problem? We need more convinced literature reviews to indicate clearly the state-of-the-art development.
For Sections 2 and 3, authors should indicate clearly why ABC is suitable for BPNN to determine BPNN’s parameters. In addition, authors should also introduce their proposed research framework more effective, i.e., some essential brief explanation vis-à-vis the text with a total research flowchart or framework diagram for each proposed algorithm to indicate how these employed models are working to receive the experimental results. It is difficult to understand how the proposed approaches are working.
For Sections 4 and 5, authors should use more alternative models as the benchmarking models, authors should also conduct some statistical test to ensure the superiority of the proposed approach, i.e., how could authors ensure that their results are superior to others? Meanwhile, authors also have to provide some insight discussion of the results. Authors can refer the following references for conducting statistical test.
Round 2
Reviewer 3 Report
Authors have completely addressed all my concerns.